# Electroacupuncture Relieves Visceral Hypersensitivity via Balancing PAR2 and PAR4 in the Descending Pain Modulatory System of Goats

**DOI:** 10.3390/brainsci13060922

**Published:** 2023-06-07

**Authors:** Panpan Guo, Qiulin Zhang, Sha Nan, Haolong Wang, Ning Ma, Faisal Ayub Kiani, Mingxing Ding, Jianguo Chen

**Affiliations:** 1College of Veterinary Medicine, Huazhong Agriculture University, Wuhan 430070, China; 2Department of Clinical Sciences, Faculty of Veterinary Sciences, Bahauddin Zakariyah University, Multan 60000, Pakistan

**Keywords:** electroacupuncture (EA), TNBS, visceral hypersensitivity, visceromotor response, protease-activated receptor 2 (PAR2), protease-activated receptor 4 (PAR4), PAG-RVM-SCDH axis

## Abstract

Electroacupuncture (EA) is an efficient treatment for visceral hypersensitivity (VH). However, the mechanism underlying VH remains obscure. This study aimed to examine the effect of EA at Housanli acupoint on PAR2 and PAR4 expression in the periaqueductal gray (PAG), rostral ventromedial medulla (RVM), and spinal cord dorsal horn (SCDH) axes, as well as on expression of the proinflammatory cytokines IL-1β and TNF-α, COX-2 enzyme, c-Fos, and the neuropeptides CGRP and SP in the same areas of the descending pain modulatory system. To induce VH in male goats, a 2,4,6-trinitrobenzene-sulfonic acid (TNBS)–ethanol solution was administered to the ileal wall. The visceromotor response (VMR) and nociceptive response at different colorectal distension pressures were measured to evaluate VH. Goats in the TNBS group displayed significantly increased VMR and nociceptive response scores, and elevated protein and mRNA levels of PAR2 and PAR4 in the descending pain modulatory system compared to those in the control group. EA alleviated VMR and nociceptive responses, decreased the protein and mRNA expression levels of PAR2, and elevated those of PAR4 in the descending pain modulatory system. EA may relieve VH by reducing PAR2 expression and increasing PAR4 expression in the descending pain modulatory system.

## 1. Introduction

Visceral hypersensitivity (VH) is defined as a low pain threshold in response to stimuli of the visceral organs and is the most debilitating and irritating symptom of inflammatory bowel syndrome (IBS). According to reports, most of the patients with IBS are afflicted with VH [1]. Costing China an estimated ¥123 billion, VH imposes a significant burden on the families of patients and society [2]. Thus, it is of utmost importance to investigate the mechanisms underlying the onset of VH and to develop effective and affordable treatments.

The mechanism of VH is associated with peripheral and central sensitizations. The production of inflammatory mediators (such as prostaglandins, serotonin, and histamine) in injured tissues causes sensitization of primary afferent neurons [3]. Peripheral sensitization can trigger neurogenic inflammation, which promotes the production of neuropeptides (SP and CGRP) and further sensitizes afferent neurons [4,5]. The peripheral sensitization signal is then transduced to the brain via ascending pathways composed primarily of vagal and spinal afferent nerves, thus inducing central sensitization [6,7]. A marker of neuronal activation, c-Fos is upregulated when neurons are activated [8]. With recurrent stimuli from the visceral organs, the nervous system maintains a state of sensitization.

Protease-activated receptors (PARs) are members of the G protein-coupled receptor family. PAR2 and PAR4 are two subtypes activated by proteolytic cleavage to unmask different N-terminal extracellular domains. Both are abundantly expressed in epithelial, brain, neuronal, and glial cells, and are related to VH [9]. PAR2 is activated by trypsin, tryptase, and cathepsin S via proteolytic cleavage [10]. It is located on resident mast cells and other inflammatory cells. When activated, it triggers the release of inflammatory mediators [11]. PAR2 is co-expressed with substance P (SP) and calcitonin gene-related peptides (CGRP) in neurons, all of which are released when inflammatory mediators sensitize peripheral afferent neurons to induce VH [12]. Shah et al. demonstrated PAR2 expression in the spinal cord was elevated in VH induced by injection of TNBS into the ileal wall of goats [13]. The fecal supernatant from patients with IBS, which induces VH in experimental animals, is incapable of sensitizing neurons from PAR2-deficient mice; thus, VH initiated by activated PAR2 is dramatically diminished in PAR2-deficient mice [14,15]. Coelho et al. revealed that sub-inflammatory doses of PAR2 agonists administered intracolonically to rats induced long-lasting visceral hyperalgesia and up-regulated c-Fos protein expression in spinal neurons [16]. Hence, PAR2 activation triggers neuroinflammation through the production of SP and CGRP and sensitizes afferent neurons to induce VH.

PAR4 is another G protein-coupled receptor implicated in visceral hyperalgesia and hypersensitivity. Thrombin, trypsin, and cathepsin G cleave PAR4 to expose its N-terminal ligand [17]. Unlike PAR2 activation, PAR4 activation results in an anti-nociceptive response. Earlier studies demonstrated that activation of PAR4 by the PAR4 agonist peptide can suppress colonic hyperalgesia by inhibiting the excitability of colonic sensory neurons. Compared to wild-type mice, PAR4-deficient mice exhibit significantly increased visceromotor responses to colorectal distension (CRD) following intracolonic administration of mustard oil [18]. Furthermore, Augé et al. showed that PAR4 co-expresses with PAR2 and TRPV4 in dorsal root ganglia. PAR4 agonist administration can significantly inhibit hyperalgesia caused by PAR2 or TRPV4 in response to colonic distension [18]. Annaházi et al. infused fecal supernatants from patients with IBS-D and ulcerative colitis (UC) into the colons of mice, respectively, or intracolonic PAR4-activating peptide or cathepsin G. The UC fecal supernatant resulted in colonic hyposensitivity to distension, similar to the results from the PAR4-activating peptide [19]. Additionally, Annaházi et al. showed that intracolonically administered PAR4 antagonist greatly exacerbated VH induced by TNBS [20]. 

Pharmacological treatments for VH include analgesics, steroidal anti-inflammatory medications, and opioids. However, their application is limited owing to severe side effects, such as addiction, intolerance to analgesics, and gastrointestinal (GI) problems. Electroacupuncture (EA) is widely used to alleviate pain, especially visceral pain associated with gastrointestinal disorders [21,22]. Recent studies established that EA mediates analgesia by activating the descending pain inhibitory system or inhibiting the descending pain facilitatory system, which is composed of the periaqueductal grey (PAG), rostral ventromedial medulla (RVM), and spinal cord dorsal horn (SCDH) [23]. Wan et al. found that EA increased the pain threshold in VH-induced goats by activating the descending pain inhibitory system [23]. A previous study observed that EA elevated the threshold of abdominal contraction in colon-distended VH-induced goats by downregulating PAR2 in the spinal cord [13]. According to traditional Chinese medicine theory, acupuncture reconciles yin and yang to treat disorders and produces an analgesic effect by regulating the body’s internal equilibrium, or the balance between proinflammatory and anti-inflammatory responses. Thus, we hypothesize that EA alleviates VH by balancing PAR2 and PAR4 (downregulating PAR2 and upregulating PAR4) in the descending pain modulatory system.

In the present study, VH was induced through injecting TNBS into the ileal wall of goats. EA was applied to goats with ileitis-induced VH to confirm whether EA relieved VH via balancing PAR2 and PAR4 expression in the descending pain modulatory system. Establishment of the VH model was confirmed by abdominal electromyography (EMG) and nociceptive response to CRD. The mRNA and protein levels of PAR2 and PAR4 were measured with qRT-PCR and Western blotting, respectively. The distribution of PAR2 and PAR4 in the descending pain modulatory system was observed with immunofluorescence. Our work may contribute to further elucidating the mechanisms by which EA relieves VH. 

## 2. Materials and Methods

### 2.1. Animals and Grouping

Thirty-six clinically healthy male goats, approximately 1 year old and weighing 22 ± 4 kg, were purchased from the Hubei Agricultural Academy of Science (Wuhan, China). Female goats were excluded from this experiment to eliminate variations in sensitivity caused by the oestrus cycle. Animal experiments were conducted according to the Regulations for the Administration of Affairs Concerning Experimental Animals of the People’s Republic of China. All protocols were approved by the Laboratory Animal Research Center of Hubei and the ethical council of Huazhong Agricultural University (permit number: HZAUGO-2020-003). The goats were subjected to binding training (1 h/day) during feeding to acclimate them to being approached by humans and lessen their stress responses. The goats were randomly divided into control (CON; *n* = 12), electroacupuncture (EA; *n* = 6), TNBS (TNBS; *n* = 12), and TNBS–electroacupuncture (TNBS + EA; *n* = 6) groups.

### 2.2. Induction of Visceral Hypersensitivity

The experimental animal model of VH was established as previously described [24]. Before anesthesia, goats were pre-dosed with intramuscular atropine sulfate (0.03 mg/kg IM; China Otsuka Pharmaceutical Co., Ltd., Tianjin, China) and xylazine-HCl (0.15 mg/kg IV; Enhua Pharmaceutical Group Co., Ltd., Xuzhou, China). Thereafter, a right flank laparotomy was conducted to expose the ileum onto sterile gauze. For goats in the TNBS and TNBS + EA groups, 1.2 mL TNBS–ethanol solution (30 mg TNBS in 40% ethanol) was injected into the ileal wall at 5 points approximately 15 cm from the ileocecal ligament using a 30 G needle attached to a 2 cc syringe.

Meanwhile, goats in the CON group were injected with an equivalent volume of saline using the same technique. To facilitate identification during sampling, both ends of the injection sites were marked with a loose silk ligature (3–0) in the mesentery. The ileum was then restored to the abdominal cavity, which was closed using the two-layer method. The status of the surgery was evaluated at least twice daily, and an iodophor and erythromycin ointment were applied to the wounds daily until they healed. Throughout recovery, each goat was intramuscularly administered tramadol hydrochloride (4 mg/kg, IM; Huazhong Pharmaceutical Co., Ltd., Xiangyang, China) to alleviate pain. The experimental design is depicted in Figure 1A.

### 2.3. Electroacupuncture

A pair of sterile needles were placed at the bilateral Housanli sites of goats in the EA and TNBS+EA groups 7 days after surgery. EA timing and intervals were set as described previously by Wan et al. [23]. Previous studies debated the anatomical location of the Housanli point [25]. Both inserted needles were connected to a WQ-6F Electronic Acupunctoscope via wires (Beijing Zhongyan Taihe Medical Instrument Co., Ltd., Beijing, China), after which a stimulation frequency of 60 Hz was applied for 0.5 h to goats immobilized in sternal recumbency. The goats in the CON and TNBS groups were restricted for the same amount of time (0.5 h) as those in the TNBS+EA and EA groups. The initial EA treatment was administered on day 7 and repeated at days 10, 13, 16, 19, 22, and 25.

### 2.4. Visceromotor Response to Colorectal Distension

This study used EMG and nociceptive response scores to characterize VMRs to varying CRD pressures. The EMG was recorded 30 min after each EA treatment. Briefly, a 12 cm polyethylene balloon lubricated with paraffin jelly was inserted 10 cm from the anus of the goats into the colon’s distal end. Thereafter, a pair of nickel steel needles 0.25 mm in diameter and 25 mm in length was inserted into the abdominal musculature, approximately 2–3 cm apart, before CRD and EMG were recorded. Afterward, the two electrodes were clamped on the skin and connected to the EMG equipment (Nanjing Tingze Medical Science and Technology Development Co. Ltd., Nanjing, China). Next, a vacuum pump was used to inflate the balloon to 20, 40, 60, 80, and 100 mmHg for 6 s each. Simultaneously, the EMG was recorded for 6 s while each pressure was held, followed by a 3 min interval before the next round of distention, for a total of three sets. The area under curve (AUC) was calculated from EMG data with MedLabV6.3 software (Nanjing medease science and technology Co., Ltd., Nanjing, China) and is reported in millivolts multiplying second (mV*s).

Nociceptive response scores were calculated based on behavioral responses to pain following CRD. The scoring procedure was based on that described by Janyaro et al. [26] (Appendix A). Two observers blinded to the experimental design observed the nociceptive responses to CRD at varying distension pressures.

### 2.5. Sample Collection

Following the last EMG recording, the goats were euthanized with 3 mg/kg xylidine thiazole. The previously marked ileum and spinal cord at the 11th thoracic vertebra were excised and divided into two segments. The upper part of the ileum was fixed in 4% buffered formaldehyde and processed for hematoxylin eosin (HE) staining. The heads of goats were put onto stereotaxic equipment, and the brain tissue was obtained for RT-qPCR, Western blotting (WB) and immunofluorescence (IF) analysis of gene and protein expression in the descending pain modulatory system (containing the ventrolateral PAG, RVM, and SCDH). According to a procedure described by Wan et al. [23], the locations and samples of the nuclei were harvested. The brain was separated into three blocks (B1–B3) [25,27]. The zero reference point for anterior–posterior coordinates was the vertical interaural plane. The horizontal zero planes (H0) were the horizontal lines connecting the interaural site and a site 25 mm above the inferior orbital rim. Figure 1B shows the three blocks (B1–B3). The positions of nuclei were determined according to the brain atlas of goats [28]. 

### 2.6. Assessment of Ileal Inflammation

During the experiment, various symptoms were monitored, including water and food intake, mortality rate, stool consistency, and presence of bloody feces. The body weight of the subjects was measured on days 0, 7, and 22. On day 7 of the experiment, six goats from the saline group and six from the TNBS group underwent a laparotomy procedure. A 6 cm section of the terminal ileum was removed from each goat, 15 cm proximal to the ileocecal junction. The excised sections were flushed with phosphate-buffered saline to detect any macroscopic changes. Two independent observers, who were unaware of the treatments, scored the macroscopic lesions on a scale of 0–10 as described by Tahir et al. [24] (Appendix A).

Ileal tissues were fixed in 4% formaldehyde and embedded in paraffin, after which the blocks were sectioned to 5 μm and mounted on slides coated with poly-L-lysine. Next, three serial slides were deparaffinized with xylene, rehydrated with a serial dilution of ethanol, and stained with HE. Optical images were captured using a light microscope (Nikon ECLIPSE 80I; Nikon, Tokyo, Japan) with 10× and 40× objective lenses. As before, two researchers evaluated the microscopic alterations blindly, using a scoring scale ranging from 0 to 10 [24] (Appendix A).

A portion of ileal tissue stored at −80 °C was ground and homogenized in 1 mL of PBS with a pH of 7.25. Afterward, the solution was centrifuged for 10 min at 5000 g and 4 °C. A BCA protein assay kit (BL521A; Biosharp, Guangzhou, China) was used to determine the supernatant protein content of the supernatant. Using matching ELISA kits (MPO: JM-00393S2, Jiangsu Jingmei Biological Technology Co., Ltd., Yancheng, China; IL-1β: JM-07720S2, Jiangsu Jingmei Biological Technology Co., Ltd., Yancheng, China; TNF-α: JM-00367S1, Jiangsu Jingmei Biological Technology Co., Ltd., Yancheng, China), the concentrations of MPO, IL-1β, and TNF-α in the ileum were determined. 

### 2.7. Western Blot Analysis

The T11 spinal cord, PAG, and RVM were ground in a mortar containing a bit of liquid N_2_ after being weighed, which was followed by protein extraction using a mixed solution of radio immunoprecipitation assay lysis buffer (RIPA, P0013C; Beyotime, Shanghai, China), phenylmethanesulfonyl fluoride (PMSF, ST506; Beyotime, Shanghai, China), and phosphatase inhibitor (P1081; Beyotime, Shanghai, China). Protein concentration was determined using a BCA protein assay kit (BL521A; Biosharp, Guangzhou, China). Each lane contained 30 g of protein, which was separated by 7.5, 10, or 15% SDS-PAG gel based on their molecular weight, transferred to PVDF membranes (PVDF, ISEQ00010; Merck, Kenilworth, NJ, USA), and blocked in 5% BSA solution for 2 h at room temperature. The membranes were then incubated with their respective primary antibodies, rabbit anti-PAR2 (ABclonal, A8796; Wuhan, China; 1:1000 diluted in 5% BSA), rabbit anti-PAR4 (Abcam, ab137927; Cambridge, MA, USA; 1:1000 diluted in 5% BSA), rabbit anti-substance P (ABclonal, A20772; Wuhan, China; 1:500 diluted in 5% BSA), rabbit anti-c-Fos (ABclonal, A0236; Wuhan, China; 1:500 diluted in 5% BSA), rabbit anti-CGRP (Abcam, AB283568; Cambridge, MA, USA; 1:1000 diluted in 5% BSA), and rabbit anti-GAPDH (ABclonal, AC001; Wuhan, China;1:3000 diluted in 5% BSA). Afterward, the membranes were rinsed with TBST solution and treated with secondary antibodies conjugated with horse radish peroxidase (donkey anti-rabbit-IgG; ABclonal, AS038; Wuhan, China; 1:10,000 in TBST) for 2 h at 30 °C. The antigen–antibody complex was observed using an ImageQuant™ LAS 4000 mini CCD camera (GE HealthCare, Chicago, IL, USA) and a horseradish peroxidase substrate (Millipore). The bands were analyzed using Quantity One software (Bio-Rad Laboratories, Hercules, CA, USA), and the results are expressed as the immunoreactivity ratio of the target gene to GAPDH.

### 2.8. Measurement of Inflammatory Mediators in the Descending Pain Modulatory System

The spinal cord and nuclei were ground and homogenized in 1 mL PBS at pH 7.25 and 4 ℃, after which the solution was centrifuged at 5000× *g* at 4 °C for 10 min. The protein concentration of the supernatant was determined using a BCA protein assay kit (BL521A; Biosharp, Guangzhou, China). The concentrations of TNF-α, IL-6, and COX-2 in the spinal cord and nuclei were assayed using the corresponding ELISA kits (TNF-α; JM-00367S1, Jiangsu Jingmei Biological Technology Co., Ltd., Yancheng, China; IL-6; JM-00403S1, Jiangsu Jingmei Biological Technology Co., Ltd., Yancheng, China; COX-2; and JM-08062S1, Jiangsu Jingmei Biological Technology Co., Ltd., Yancheng, China) according to the manufacturer’s instructions. 

### 2.9. Immunohistochemistry and Immunofluorescence

T11 spinal cord and brain sections were routinely immersed in paraffin with the front end of the transverse section facing up, sectioned to 5 μm using a microtome, placed on glass slides coated with polylysine, deparaffinized, and rehydrated sequentially. All slices were preincubated for 1 h at 37 °C with 10% normal donkey serum. Subsequently, each of the three sections were incubated overnight in a moist chamber with rabbit anti-PAR2 (ABclonal, A8796; Wuhan, China; 1:200 dilution in PBS) or rabbit anti-PAR4 (Abcam, ab137927; Cambridge, MA, USA; 1:200 dilution in PBS), and mixed with mouse anti-NeuN (Proteintech Group, 66836-1-lg; Rosemont, IL, USA; 1:200 dilution in PBS) or mouse anti-GFAP (Cell Signaling Technology, 3670S; Danvers, MA, USA; 1:200 dilution in PBS). As a negative control, the remaining slides were treated with PBS instead of the appropriate antibody. Next, the slides were washed with PBS solution (pH = 7.35–7.45) and incubated with CY3-labeled donkey anti-rabbit antibody (Servicebio, GB114304; Wuhan, China; 1:200 dilution in PBS) at 37 °C for 1 h. The slides were then washed with PBS, incubated with FITC-labeled goat anti-mouse antibody (Servicebio, GB121152; Wuhan, China; 1:200 dilution in PBS), rinsed with PBS solution, and incubated for 5 min at room temperature with DAPI (Beyotime, C1002; Shanghai, China, 1:50,000 diluted in PBS;). Optical images were taken using a 20x objective fluorescent microscope (Nikon). Red fluorescence indicates PAR2 or PAR4, green fluorescence indicates neurons or astrocytes, blue fluorescence indicates the cell nucleus, and yellow fluorescence indicates colocalization of PAR2 +NeuN or GFAP and PAR4 +NeuN or GFAP. A quantitative analysis of signal intensity was performed using ImageJ software (NIH, Bethesda, MD, USA).

T11 spinal cord and brain sections fixed in 10% neutral-buffered formalin were routinely embedded in paraffin with the front end of the transverse section facing up. Ten serial sections of each area were sectioned to 5 μm using a microtome and mounted on glass slides coated with polylysine, deparaffinized, and rehydrated sequentially. All slices were preincubated for 1 h at 37 °C with 10% normal donkey serum. Three of the ten slides were incubated with rabbit anti-c-Fos IgG (Abmart, TA0132S; Shanghai, China; 1:500 diluted in PBS), rabbit anti-substance P IgG (Abmart, TD7522; Shanghai, China; 1:500 diluted in PBS), and rabbit anti-CGRP IgG (Abcam, AB283568; Cambridge, MA, USA; 1:500 diluted in PBS). As a negative control, the remaining slide of each area was treated with PBS instead of the primary antibody. The experimental procedures for SABC immunohistochemistry adhered to the reagent manufacturer’s protocols (Boster Bio, Wuhan, China). Positive cells have their cytoplasm or nucleus colored yellow or dark brown.

We obtained optical images of the stained cells using a light microscope (Nikon ECLIPSE 80I; Nikon) and a computer- and video-based system (High-Resolution Pathological Image Analysis System 1000; Wuhan Qianping Ltd., Wuhan, China). Three slides of each type were examined using a 40× objective lens. Image-Pro Plus 6.0 (Media Cybernetics, Rockville, MD, USA) was used to evaluate the optical density of immunoreactive c-Fos, SP, and CGRP staining.

### 2.10. Real-Time Quantitative PCR

The total RNA from the spinal cords, PAGs, and RVMs of each group was isolated using Trizol reagent (Invitrogen, Waltham, MA, USA). Next, cDNA was generated using a FIRST Strand cDNA Synthesis Kit and 900 ng of total RNA (TOYOBO, Osaka, Japan). Table 1 displays the primer sequences for PAR-2, PAR-4, SP, CGRP, c-Fos, and GAPDH. Next, qPCR was conducted using a Step One PlusTM Real-Time PCR System (Applied Biosystems, Waltham, MA, USA) with a SYBER Green RT-PCR kit (Takara Bio, Shiga, Japan). PAR-2, PAR-4, SP, CGRP, and c-Fos mRNA levels were measured using the 2^−ΔΔCt^ method, where ΔCt = Ct_target gene_ − Ct_GAPDH_.

### 2.11. Correlation Analysis

The ratio of PAR2/PAR4 expression was computed by dividing the relative expression of PAR2 by that of PAR4. Correlation analysis between the expression of PAR2, PAR4, and their expression ratio was then conducted using GraphPad Prism 9.0 (GraphPad Software, Boston, MD, USA), and the results were illustrated using a heatmap.

### 2.12. Statistical Analysis

All data are expressed as mean ± SD. The VMR data were analyzed with GraphPad Prism v9.0 (GraphPad Software Inc., Boston, MD, USA) using two-way ANOVA followed by Bonferroni’s post hoc test. The statistical comparisons for parametric data (body weight changes, IF, RT-PCR, Western blot, and ELISA) were carried out with GraphPad Prism v9.0 using one-way analysis of variance (ANOVA) followed by Bonferroni post hoc test. The statistical differences of non-parametric values (macroscopic scores, microscopic scores, and nociceptive response scores) among groups were identified with SPSS version 18.0 (SPSS Inc., Chicago, IL, USA) using Kruskal–Wallis ANOVA followed by the rank-based Mann–Whitney U test. A difference was considered significant if the *p*-value was less than 0.05. Correlations and significant difference between the expression levels of two receptors and their expression ratio with VMR were determined by Pearson correlation analysis.

## 3. Results

### 3.1. TNBS-Induced Ileal Inflammation

TNBS-treated goats exhibited a marked aversion to food, lethargy, diarrhea, and significant weight loss on day 7. Compared with the saline-treated goats, TNBS-treated goats showed significantly decreased body weight on day 7 (*p* < 0.05), and at day 25, TNBS-treated goats showed lower body weight than the goats in the other three groups. Goats in the TNBS + EA group had slightly lower body weights than goats in the CON and EA groups (*p* > 0.05). No significant difference was observed in the body weight between the goats in EA and CON group (*p* > 0.05) (Figure 2C).

No significant histopathological changes were observed in the saline-treated ileum on days 7 and 25. Macroscopic studies of the TNBS-treated ileum on day 7 revealed severe ileitis with intermittent pseudo-membrane adhesion, thickening, ulceration, and necrosis. Furthermore, TNBS injected into the ileal wall did not appear to harm the organs or tissues near the administration site (Figure 2A). At the macroscopic level, the TNBS-treated ileum displayed more severe lesions (*p* < 0.05) than did the saline-treated ileum at day 7. The ileum treated with TNBS recovered progressively and did not differ significantly from the other groups at the macroscopic level on day 25 (Figure 2B). On day 7, the concentrations of MPO, IL-1β, and TNF-α in goats treated with TNBS were significantly higher (*p* < 0.05) than those in goats treated with saline. At day 25, the concentrations showed no significant difference among the four groups (Figure 2D).

The lamina propria were extensively infiltrated by neutrophils and lymphocytes, resulting in ulceration and blood vessel congestion extending into the submucosal and muscular layers in the ileal wall of goats treated with TNBS on day 7. Mild or moderate infiltration of inflammatory cells was noted in the submucosal and muscular layers on day 25 (Figure 2E). Lesions in the ileal walls of TNBS-treated goats were more severe (*p* < 0.05) than those in goats of the CON group on day 7. On day 25, the ileal walls of goats treated with TNBS also exhibited more severe (*p* < 0.05) lesions than those in the other three groups, while there were no differences in the microscopic lesions among goats (*p* > 0.05) in the CON, EA, and TNBS + EA groups (Figure 2F).

### 3.2. Effect of Repeated Electroacupuncture Treatment on Response to Colorectal Distension

The EMG of the abdominal muscles used for the evaluation of VH is depicted in Figure 3A–H, and the VMR to the graded CRD pressure (20–100 mmHg) was recorded. As the CRD pressure increased, the VMR increased. Compared with goats in the CON group, those in the TNBS group exhibited a higher (*p* < 0.05) VMR in response to 20–100 mmHg distension pressures on days 7–25. In contrast to those in the CON group, goats in the EA group exhibited a decreased VMR (*p* < 0.05) in response to 100 mmHg distension pressure on day 25, such that there was no significant decrease in VMR in response to 20–100 mmHg distension pressures on days 7–22 (*p* > 0.05). Compared with the goats in the TNBS group, those in the TNBS + EA group exhibited decreased VMR (*p* > 0.05) in response to 20–100 mmHg on day 7 and significantly reduced VMR (*p* < 0.05) to 20–100 mmHg distension pressures on days 10–25.

When various distension pressures were applied, the experimental goats displayed various symptoms, including fast breathing, tail wagging, guarding, restlessness, lip curling, neck movement, and postural change. Compared with goats in the CON group, TNBS-treated goats experienced significantly enhanced (*p* < 0.05) behavioral reactions to 40–100 mmHg distension pressure on days 7–25. On days 7–25, goats treated with EA exhibited significantly fewer behavioral reactions (*p* < 0.05) to 40–100 mmHg compared to those in the CON group. Compared with goats in the TNBS group, those in the TNBS+EA group decreased behavioral reactions to 40–100 mmHg on days 7–25(*p* < 0.05) (Figure 4A–G).

Compared with levels in the CON group, the concentrations of TNF-α, IL-1β, and COX-2 in the descending pain modulatory system were significantly increased in the TNBS group (*p* < 0.05), while no difference was observed in the EA group (*p* > 0.05). Moreover, in the TNBS + EA group, the concentrations of TNF-α, IL-1β, and COX-2 in the three nuclei and areas of the descending pain modulatory system were significantly decreased compared with those in the TNBS group (*p* < 0.05) (Figure 5).

### 3.3. Effect of Electroacupuncture on PAR2 and PAR4 Expression in the Descending Pain Modulatory System

PAR2 and PAR4 expression levels are shown in Figure 6A1,E1,I1. PAR2 was expressed similarly in PAG, RVM, and SCDH. PAR2 protein expression decreased significantly in the EA group compared with that in the CON group (*p* < 0.05). Furthermore, its expression increased significantly (*p* < 0.05) in the TNBS group compared with levels in the TNBS + EA and CON group. PAR4 expression levels in the PAG, RVM, and SCDH also showed similar changes in these areas. PAR4 protein expression was significantly elevated in EA-treated goats compared with CON goats (*p* < 0.05). There was no significant difference in PAR4 expression between the TNBS and CON groups (*p* > 0.05). Goats treated with TNBS + EA exhibited significantly increased PAR4 expression compared to goats in the TNBS group (*p* < 0.05). Relative PAR2 and PAR4 mRNA expression levels were consistent with their respective protein expression levels (Figure 6A2,E2,I2).

PAR2 and PAR4 were predominantly colocalized with the neuronal marker NeuN and to a lesser extent with the marker of astrocyte activation, GFAP, in the PAG, RVM, and SCDH (Figure 7 and Figure 8). The changes in the immunofluorescence intensities of PAR2 and PAR4 were consistent with the changes in their expression levels. The results are shown in Figure 9. 

We performed a correlation analysis between PAR2 and PAR4 expression levels and the expression ratio of PAR2/PAR4 with VMR in the descending pain modulatory system from the CON, EA, TNBS, and TNBS + EA groups. To better reflect the correlation between the three values and VMR before and after EA, we analyzed the correlation between the three values and VMR in the CON and EA groups, and TNBS and TNBS + EA groups, respectively. The correlation analysis results of the CON and EA groups are shown in Figure 6C,G,K, while those of the TNBS and TNBS + EA groups are shown in Figure 6D,H,L. The expression ratios of PAR2/PAR4 in the PAG, RVM, and SCDH were significantly lower (*p* < 0.05) in goats from the EA group than in those from the CON group. This ratio was also significantly lower (*p* < 0.05) in goats from the TNBS + EA group than in those from the TNBS group. The PAR2 protein expression level and the ratio of PAR2/PAR4 showed a significant positive correlation with VMR in the TNBS and TNBS + EA groups, whereas the PAR4 expression level was significantly negatively correlated with VMR. 

### 3.4. Effect of Electroacupuncture on c-Fos, CGRP, and SP Expression in the Descending Pain Modulatory System

Changes in the expression levels of c-Fos, CGRP, and SP in the PAG, RVM, and SCDH are shown in Figure 10A1, B1, C1. No significant differences (*p* > 0.05) were observed in the expression of CGRP or SP in the descending pain modulatory system between the goats in the CON group and EA group. The expression of c-Fos in the descending pain modulatory system was significantly increased (*p* < 0.05) in goats of the EA group, compared with levels in the CON group. In contrast to levels in the CON group, c-Fos, SP, and CGRP expression levels were significantly upregulated in the three nuclei and areas of the descending pain modulatory system in the TNBS group (*p* < 0.05), and this regulation change was significantly reversed (*p* < 0.05) by EA. The changes in SP and CGRP mRNA expressions were consistent with the changes in their protein expressions (Figure 10A2, B2, C2). While there were no significant differences between c-Fos mRNA expression levels of the CON and EA groups (*p* > 0.05), c-Fos mRNA expression was increased (*p* < 0.05) in the TNBS group compared to levels in the CON and TNBS + EA groups.

Immunohistochemical staining of sections from the PAG, RVM, and SCDH in the four groups showed variations in distributions of immunoreactivities of c-Fos, CGRP, and SP (Appendix A). Goats in the TNBS group showed enhanced c-Fos, CGRP, and SP immunoreactivity compared with those in the CON group. However, c-Fos, CGRP, and SP immunoreactivities were lower in the TNBS+EA group than in the TNBS group. CGRP and SP immunoreactivities were lower in the EA group than in the CON group, while c-Fos showed enhanced immunoreactivity in the EA group than in the CON group. 

## 4. Discussion

Recently, there was growing interest in VH; however, the mechanisms underlying its pathogenesis remain unknown. Multiple animal models of VH were used to explore these mechanisms. Some studies reported that repeated CRD in rat neonates induces VH in adulthood [29,30]. Lee et al. examined rats with water avoidance stress-induced colonic hypersensitivity [31]. Lopez-Estevez et al. showed that dextran sulfate sodium administration induced VH in animal models [32]. A previous study reported that administering TNBS into the ileal lumens of rats induced ileitis-derived VH [33]. Janyaro et al. [26] constructed VH models by administering TNBS to the ileal walls of goats. In our study, goats administered a TNBS–ethanol solution (1.2 mL; 30 mg TNBS dissolved in 40% ethanol) into their ileal walls via laparotomy-exhibited gastrointestinal signs and symptoms such as anorexia, diarrhea, and weight loss, as well as significant pathologic changes and elevated levels of inflammatory cytokines such as MPO, IL-1β, and TNF-α on day 7 compared with goats treated with saline. The presenting symptoms and pathological abnormalities were consistent with the previous findings in our lab, and resemble the results obtained by Merritt et al. [34]. Furthermore, consistent with the findings of Tahir et al. [24] and Shah et al. [33], we found that both the VMR and nociceptive response to CRD at graded pressures progressively raised reached a maximum on day 13 and remained at a higher level at day 25. Thereafter, how long the VH persists needs to be investigated. Zhou et al. infused TNBS in 50% ethanol and an equivalent volume of 50% ethanol or saline into the colon of rats to induce VH and found that rats treated with TNBS exhibited evidence of visceral as well as somatic hypersensitivity compared to saline- and ethanol-treated rats, while rats treated with saline and ethanol showed no significant difference in VH [35]. The results obtained by our lab demonstrate that the effects of TNBS and TNBS + shamEA on VMR and nociceptive response to CRD showed no difference [13,23]. Other researchers also omitted ethanol control or TNBS + shamEA control in their experiments [10,36,37,38]. Thus, in the present study, it was reasonable to take the goats injected with saline as the control group without ethanol control and TNBS + shamEA control considering animal welfare.

Following irritation by inflammatory mediators, peripheral sensitization occurs, leading to the increased release of SP and CGRP to transmit the nociceptive signal to the central nervous system. Therefore, the overexpression of SP and CGRP in the central nervous system facilitates neuronal excitability and neuroinflammation, which contributes to the release of inflammatory mediators, resulting in their signaling back to the neurons and hence inducing a positive feedback loop of sensitization [39,40,41]. Numerous previous studies showed increased expression of inflammatory mediators (such as TNF-α, IL-6 and COX-2) in the brain following TNBS- or dextran sulfate sodium-induced colitis [42,43,44]. In this study, SP, CGRP, TNF-α, IL-6, and COX-2 levels were significantly increased in the PAG, RVM, and SCDH of goats following TNBS administration. These results indicate that TNBS induces central sensitization that contributes to neuroinflammation, thus provoking the release of inflammatory mediators; c-Fos is a marker of neuronal activity, and its overexpression indicates enhanced nervous excitability. In our study, c-Fos expression was upregulated in the TNBS group, which showed that high excitability probably existed in the descending pain facilitatory system with increased VMR to the CRD.

Several studies revealed that PAR2 activation induces VH by triggering the production of neuropeptides in humans and animals. An intracolonically administered PAR2 agonist (SLIGRL-NH_2_) can elicit sustained VH with enhanced spinal cord c-Fos expression [13]. Sub-inflammatory dosages of PAR2-AP administered intraplantar to mice elicit heat hyperalgesia, and the activation of PAR2 stimulates the release of neuropeptides (SP and CGRP) in neurons to generate neuroinflammation [45,46]. Most neurons show elevated CGRP expression following central inflammation and mice lacking CGRP expression in the central nervous system are not affected by inflammation-induced hyperalgesia [47]. In our study, TNBS treatment increased PAR2 expression and triggered neurogenic inflammation with the release of SP and CGRP in the descending pain modulatory system. Importantly, PAR2 activation positively correlated with increased VMR to CRD.

In contrast to PAR2, PAR4 exerts analgesic and antinociceptive effects. Asfaha et al. demonstrated that PAR4 agonists can raise the threshold for mechanical and thermal stimulation and reduce inflammatory hyperalgesia. Moreover, following intracolonic infusion of sub-inflammatory PAR4-AP, animals showed hyposensitivity to colorectal distension [19]. PAR4 is co-expressed with PAR2, which can strongly cause VH; intracolonic infusion of a PAR4 agonist considerably decreases VH induced by administration of a PAR2 agonist [18]. PAR2 and PAR4 co-localize with neurons in the central nervous system. The results of our study indicate that neuronal PAR2 and PAR4 are involved in the modulation of VH and that PAR2 might exert a pronociceptive effect. In contrast, the PAR4 overexpression might be connected to its role as an endogenous analgesic factor against body-produced nociception.

EA is a globally recognized pain-reducing agent. It is used to treat various pain conditions with moderate efficacy and few adverse effects [48]. Several variables, including frequency, acupoints, and intervals, determine its analgesic effects. A recent study determined that the optimal frequency for EA-induced analgesia in goats was 60 Hz. Sun et al. [49] demonstrated that bilateral EA administration at the Housanli considerably reduced VH. A previous study compared the therapeutic ability of EA to alleviate pain at varying intervals and demonstrated that EA sessions spaced 2 days apart could cause cumulative analgesia [23]. In our study, TNBS-induced VH was attenuated in goats after repeated EA at the bilateral Housanli acupoints. After five sessions of EA, the goats in the TNBS+EA group demonstrated the most significant relief from VH compared with those of the TNBS group.

EA alleviates VH mostly by modulating neurotransmitters and neuromodulators (i.e., endogenous opioids, serotonin, and norepinephrine) and by decreasing the generation of inflammatory mediators [50,51,52]. Previous studies showed that c-Fos, SP, and CGRP expression levels are markedly decreased following EA therapy [13]. Visceral nociception is associated with the descending pain modulatory system. Zhao et al. postulated that the analgesic effect of EA is mediated by activating the descending pain inhibitory system or inhibiting the descending pain facilitatory system, which is composed mostly of the PAG, RVM, and SCDH [53]. There is evidence that when chemical or electrical stimuli are applied to the PAG, the nociceptive response of SCDH neurons to CRD is reduced [54,55]. EA can interfere with the transmission of nociceptive signals from SCDH to supraspinal areas [56,57]. In our study, the decreased expression of inflammatory mediators (TNF-α, IL-6, and COX-2), SP, and CGRP in the descending pain modulatory system indicated a reduction in neuroinflammation and the alleviation of central sensitization due to EA treatment.

EA relieves VH via the opioid system [58], adenosine pathway [59], and cannabinoid system [60], and other endogenous substances participate in the modulation of VH by EA. PAR2 and PAR4 are two candidate receptors. There were few studies on the EA-induced expression of PAR2 and PAR4 in IBS models, particularly in the descending pain modulatory system in goat ileitis. In the present study, EA attenuated TNBS-induced VH by upregulating PAR4 and downregulating PAR2 mRNA and protein levels. Although the analgesic mechanism of PAR4 is unclear, it remains to be determined how EA relieves VH by increasing PAR4 expression, possibly through the endothelin type A receptor system [61]. A previous study reported that EA alleviates VH by inhibiting the expression of PAR2 [13] via the PKC-TRPV pathway [62]. Our findings indicate that EA decreases the expression of PAR2, SP, and CGRP in the descending pain modulatory system while alleviating neuroinflammation. Conversely, both mRNA and protein levels of PAR4 in the descending pain modulatory system were markedly upregulated after EA, which is related to VH remission. The expression of PAR2 and PAR4 exhibited opposite correlations with VMR after EA, and the expression ratio of PAR2/PAR4 correlated more strongly with VMR than with the expression of PAR2 and PAR4. These results indicate that EA induced the upregulation of PAR4 and the downregulation of PAR2 in descending pain modulatory systems and relieved VH by regulating PAR4 and PAR2 expression, such that PAR4 is considered an endogenous analgesic and PAR2 is a pro-analgesic.

## 5. Conclusions

In this study, we demonstrated that recurrent EA at the bilateral Housanli dramatically alleviated VH, probably by regulating the expression of PAR2 and PAR4. Although it needs to be verified that balancing PAR2 and PAR4 expression mediated acupuncture, contributing to alleviating VH, our study suggested EA increased PAR4 expression and decreased PAR2 in the descending pain modulatory system.

## Figures and Tables

**Figure 1 brainsci-13-00922-f001:**
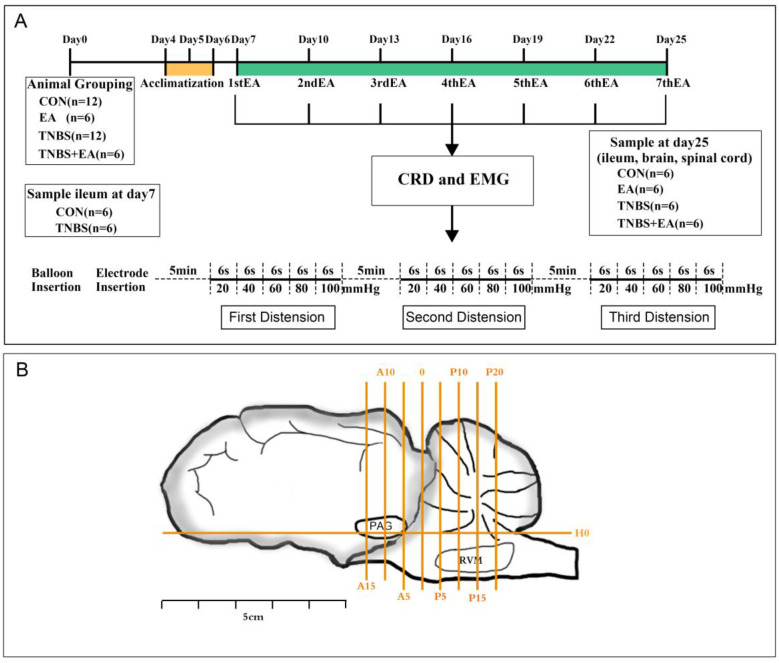
(**A**) The scheme of the experiment. CRD: colorectal distension; VMR: visceral motor response. (**B**) Brain sectioning. H0: The horizontal reference line. Transverse planes A15, A10, A5, and 0 are shown 15 mm anterior, 10 mm anterior, and 0 mm anterior to the interaural line, and those P5, P10, P15, and P20 are shown 5 mm, 10 mm, 15, and 20 mm posterior to the interaural line. Periaqueductal gray (PAG) and rostral ventromedial medulla (RVM) are the nuclei and areas identified.

**Figure 2 brainsci-13-00922-f002:**
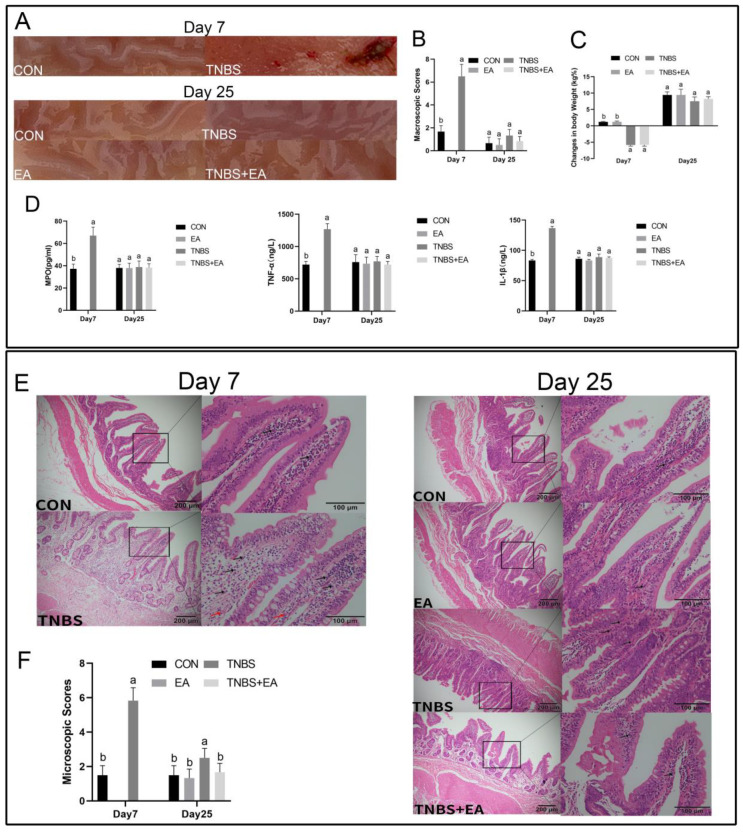
(**A**,**B**) Macroscopic pathological changes at day 7 and 25 and macroscopic change scores at day 7 (mean ± SD, *n* = 6) and day 25 (mean ± SD, *n* = 6). Kruskal–Wallis ANOVA followed by the rank-based Mann–Whitney U test. Macroscopic change scores were based on a scale of 0–10, adhesions (0–2), mucosal hyperemia (0−3), ulcer (0–3), and wall thickness (0−2). (**C**) Changes in body weight at day 7 and day 25 (mean ± SD, *n* = 6). One-way ANOVA followed by Bonferroni’s post-test. (**D**) MPO, TNF-α, and IL-1β concentrations in ileum tissue at day 7(mean ± SD, *n* = 6) and day 25 (mean ± SD, *n* = 6). One-way ANOVA followed by Bonferroni’s post-test. (**E**) Representative images of the ileum tissue stained with hematoxylin and eosin (HE) at day 7 and day 25. Red arrows show blood vessel congestion and black arrows show the inflammation cells. The bars = 200µm or 100µm. (**F**) Microscopic change scores were determined based on the crypt depth (0−2 mm), inflammatory cells (0−3), blood vessel congestion (0−3), and ulceration (0−2) at day 7, and day 22 (mean ± SD, *n* = 6). Kruskal–Wallis ANOVA followed by the rank-based Mann–Whitney U test. The values with different letters differ significantly (*p* < 0.05).

**Figure 3 brainsci-13-00922-f003:**
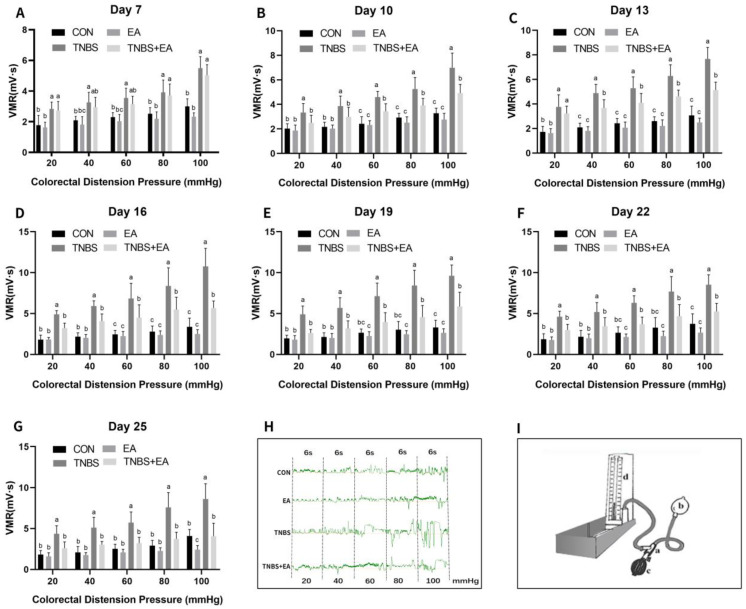
The effects of EA and TNBS on the visceromotor responses (VMR) of goats in four groups. (**A**–**G**) Effects of repeated EA treatments on VMR to colorectal distention pressure (CRD) measured with electromyography (EMG) at day 7, 10, 13, 16, 19, 22, and 25 (mean ± SD, *n* = 6). The values with different letters differ significantly (*p* < 0.05). Two-way ANOVA followed by Bonferroni’s post-test. (**H**) The representative EMG traces with 20, 40, 60, 80, and 100 mmHg distention pressures. The pressure continuously increased from 20, 40, 60, 80, to 100 mmHg by stage and lasted for 6 s at each stage. (**I**) The distension device made with a T-connector (a), connecting a balloon (b), a vacuum pump (c), and a sphygmomanometer (d).

**Figure 4 brainsci-13-00922-f004:**
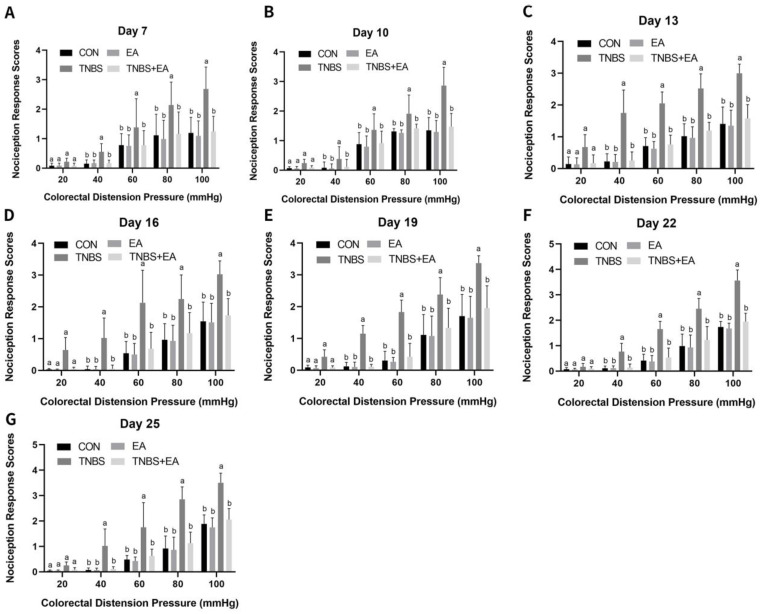
(**A**–**G**) Effects of repeated EA treatments on nociception response to CRD at day 7, 10, 13, 16, 19, and 22 (mean ± SD, *n* = 6). The nociception response scores on a 0–4 scale, normal behavior—0, slightly modified behavior—1, mild behavior—2, moderate behavior—3, and severe behavior—4. The values with different letters differ significantly (*p* < 0.05). Kruskal–Wallis ANOVA followed by the rank-based Mann–Whitney U test. Effect of electroacupuncture on inflammatory mediators in the descending pain modulatory system.

**Figure 5 brainsci-13-00922-f005:**
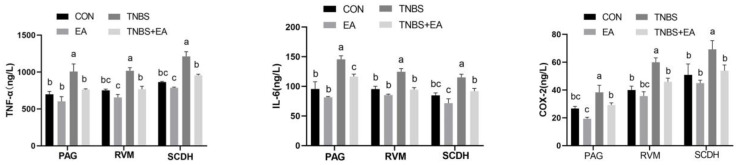
The concentrations of TNF-α, IL-6, and COX-2 in PAG, RVM, and spinal cord (SC), respectively. The values labeled with different letters indicate statistical significance (*p* < 0.05). One-way ANOVA followed by Bonferroni’s post-test.

**Figure 6 brainsci-13-00922-f006:**
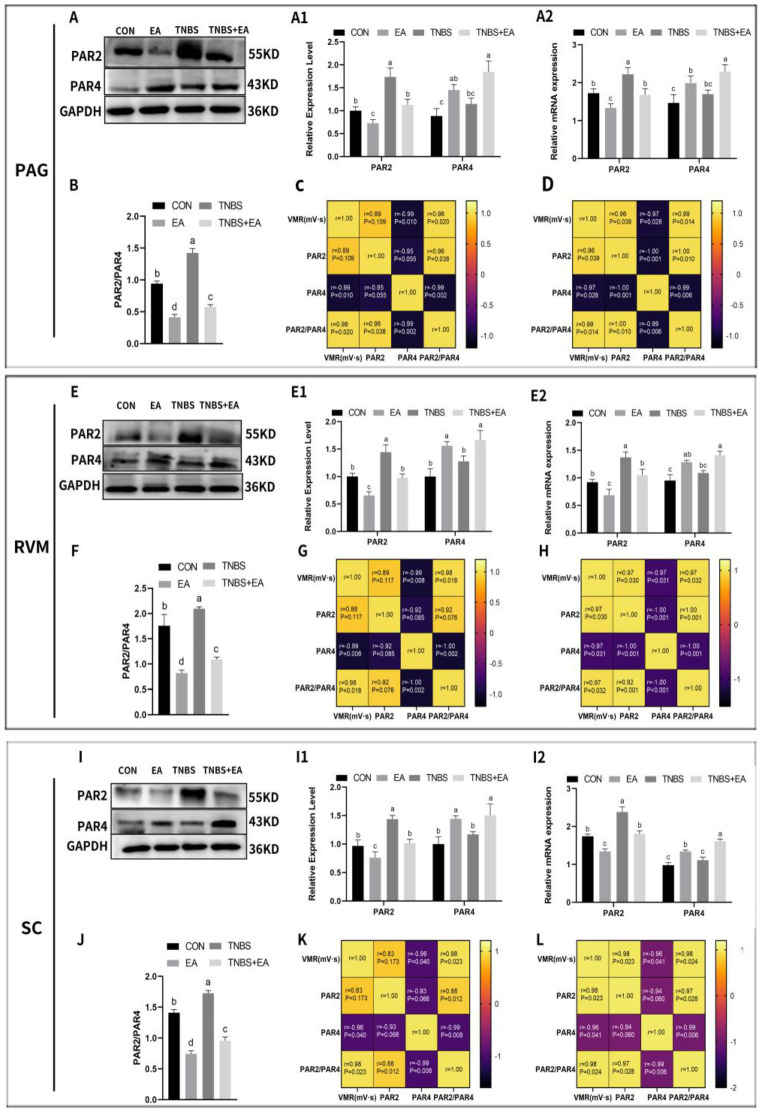
Effects of repeated EA treatments on protein expression and mRNA expression of protease-activated receptor 2 (PAR2) and protease-activated receptor 4 (PAR4) in a descending pain modulatory system (mean ± SD, *n* = 6). (**A**,**E**,**I**) The Western blotting bands of PAR2 and PAR4 in PAG, RVM, and SC, respectively. (**A1**,**E1**,**I1**) The protein expression of PAR2 and PAR4 in PAG, RVM, and SC, respectively. (**A2**,**E2**,**I2**) The mRNA expression of PAR2 and PAR4 in PAG, RVM, and SC, respectively. (**B**,**F**,**J**) The protein expression ratio of PAR2/PAR4 in PAG, RVM, and SC, respectively. (**C**,**G**,**K**) Person’s correlation (confidence interval: 95%) of VMRs and protein levels of PAR2 or PAR4, or PAR2/PAR4 ratio between the CON and EA groups in PAG, RVM, and SC, respectively, *n* = 6. (**D**,**H**,**L**) Person’s correlation (confidence interval: 95%) of VMRs and protein levels of PAR2 or PAR4, or PAR2/PAR4 ratio between the TNBS and TNBS+EA groups in PAG, RVM, and SC, respectively, *n* = 6. The values labeled with different letters indicate statistical significance (*p* < 0.05). One-way ANOVA followed by Bonferroni’s post-test.

**Figure 7 brainsci-13-00922-f007:**
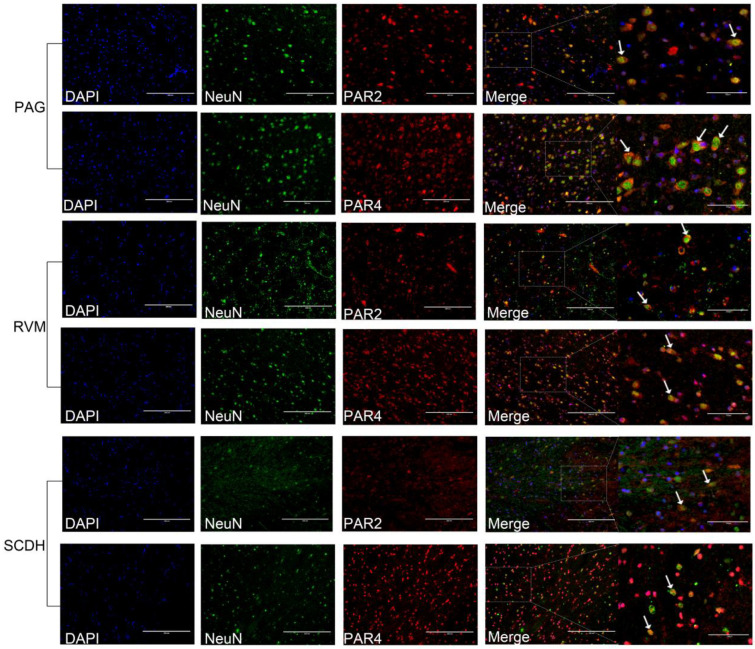
Colocalization of PAR2 and PAR4 with NeuN (the marker of neuronal activation) in PAG, RVM, and SCDH, respectively. White arrows show the colocalization of PAR2 or PAR4 with neurons. Scale bar = 200 μm or 100 μm.

**Figure 8 brainsci-13-00922-f008:**
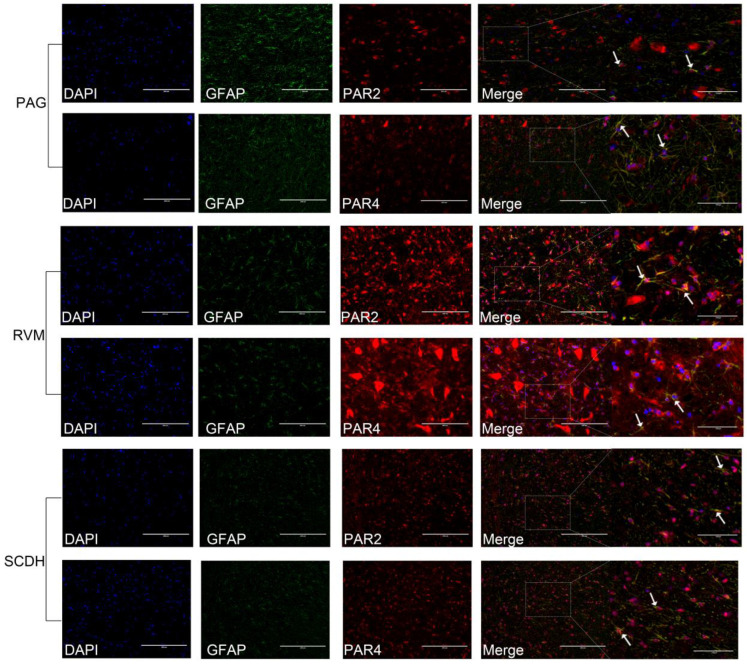
Colocalization of PAR2 and PAR4 with GFAP (the marker of astrocyte) in PAG, RVM, and SCDH, respectively. White arrows show the colocalization of PAR2 or PAR4 with astrocytes. Scale bar = 200 μm or 100 μm.

**Figure 9 brainsci-13-00922-f009:**
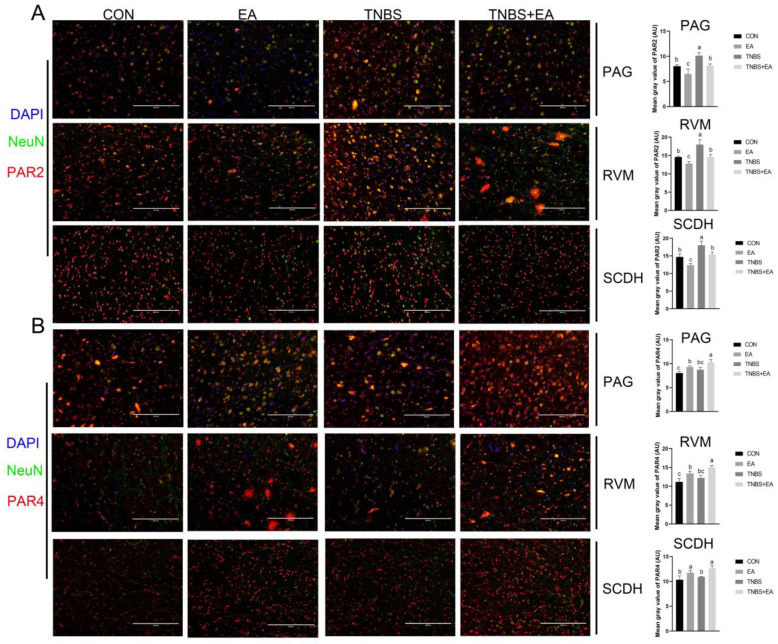
The mean immunofluorescence intensity of PAR2 and PAR4 in PAG, RVM, and SCDH in four groups (mean ± SD, *n* = 6). Scale bar = 200 μm. The values labeled with different letters indicate statistical significance (*p* < 0.05). One-way ANOVA followed by Bonferroni’s post-test.

**Figure 10 brainsci-13-00922-f010:**
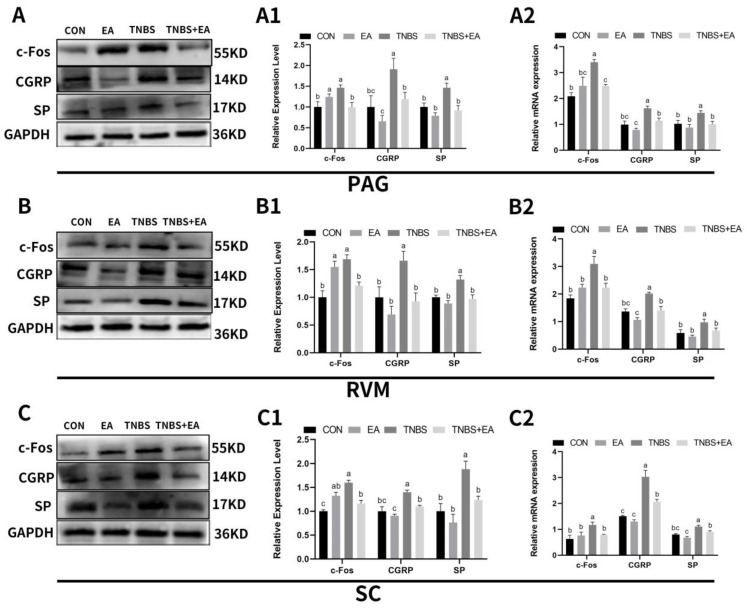
Effects of EA and TNBS on the expression of c-Fos, CGRP, and SP in the descending pain modulatory system. (**A**,**B**,**C**) The Western blotting bands of c-Fos, CGRP, and SP in PAG, RVM, and SC, respectively. (**A1**,**B1**,**C1**) The protein expression of c-Fos, CGRP, and SP in PAG, RVM, and SC, respectively. (**A2**,**B2**,**C2**) The mRNA expression of c-Fos, CGRP, and SP in PAG, RVM, and SC, respectively. The values labeled with different letters indicate statistical significance (*p* < 0.05). One-way ANOVA followed by Bonferroni’s post-test.

**Table 1 brainsci-13-00922-t001:** Primer sequences of primer sequences of PAR-2, PAR-4, SP, CGRP, c-Fos, and GAPDH.

Name	Accession Number	Primer Sequence
PAR-2	XM_018053694.1	F:5′- GATCTGCTTCACGCCCAGTA-3′
		R:5′-CCGGAAGTCCTGTGAAACGA-3′
PAR-4	XM_018051355.1	F:5′- CTGCTGCTGCACTTCTCAAAC-3′
		R:5′-ATAGATGAAGGGGTCCACGC-3′
SP	XM_005686308.3	F:5′- CAGCGACCAGATCAAGGAGG-3′
		R:5′-CATGTCCAGCATCCCGTTTG-3′
CGRP	XM_005697758.3	F:5′- GATATGGAAGTGAAGGATGCC-3′A
		R:5′-ACAATCTCAGGACTCTGGTGC-3′
c-Fos	XM_018063375.1	F:5′- CTTCAACGCCGACTACGAGG-3′
		R:5′-TCTGCCGGTGAGTGGTAGTA-3′
GAPDH	XM_005680968.3	F:5′- CTGCTGCTGCACTTCTCAAAC-3′
		R:5′-ATAGATGAAGGGGTCCACGC-3′

## Data Availability

We have provided the raw data in the Appendix A.

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
