# Peer review of "Electroacupuncture Relieves Visceral Hypersensitivity via Balancing PAR2 and PAR4 in the Descending Pain Modulatory System of Goats"

_brainsci, 2023, doi:10.3390/brainsci13060922_

Round 1
Reviewer 1 Report
Authors presented a very interesting paper about Electroacupuncture relieves visceral hypersensitivity via balancing Protease-activated receptors (PAR2 and PAR4) in the descending pain modulatory system and the role of Electroacupuncture.
This is of particular interest for quality of life of patients affected by inflammatory bowel disease and for healthcare system due to the costs to manage visceral pain.
The study was conducted on goats and seems very invasive, I did not find the NCT registration coud you please provide number?
Many thanks
Reviewer 2 Report
The reviewer would like to declare no conflict of interests with the authors and their affliations.
Title and abstract: The author should specify the species of subject employed in the study, and specify which acupoints were stimulated in this study. These information are important for the readers, as different animal models may respond to electroacupuncture differently. Furthermore, different acupoints may induce differential physiological responses.
Introduction: The authors described the previous studies on PAR2 and PAR4 in the peripheral nervous system (peripheral nerve terminals, dorsal root ganglia, etc.). SInce the main scope of the current study is on the possible contribution of electroacupuncture on PAR2 and PAR4 systems in the descending pain inhibitory pathway, the emphasis should be on the CNS. Kindly revamp this section.
Animals: The author employed goats as animal model. Kinldy justify the purpose of using such large mammals for fundamental research. Would the outcome be superior than lab rodents?
minor language edit required.
Reviewer 3 Report
Comments and suggestions are presented in the attached Word file

The manuscript should be revised by a native english speaker.
Round 2
Reviewer 3 Report
please see the attached file
